# Building Primary Preservice Teachers' Identity as Engineering Educators

Nicholas Lux [1,*] , Rebekah Hammack [1], Blake Wiehe [1] and Paul Gannon [2]

[1] Department of Education, Montana State University, Bozeman, MT 59717, USA
[2] Chemical & Biological Engineering Department, Montana State University, Bozeman, MT 59717, USA
* Correspondence: nicholas.lux@montana.edu

**Abstract:** The purpose of this qualitative case study was to investigate how two primary preservice teachers built their engineering education identities during a clinical field experience that emphasized engineering education. More specifically, we explored the development of their engineering education identities while facing unforeseen circumstances and unfamiliar engineering content. We used a nested qualitative case study approach that was bounded by a university practicum field experience that took place at the height of the COVID-19 pandemic. Data sources included preservice teacher interviews and reflective field notes. We found that the preservice teachers faced a series of contextual factors in the clinical experience that both afforded and constrained professional learning opportunities that influenced their identity development. The affordances made professional learning opportunities possible, while the constraints limited professional growth. We also found that it was the negotiation of the factors, where the preservice teachers worked to mitigate the effect of the constraints while maximizing the advantages of the affordances, that had the greatest influence on their engineering pedagogical knowledge and engineering teaching self-efficacy. Findings from this study could provide teacher educators with insight into preparing primary teachers for unexpected challenges when teaching engineering, as well as how to best prepare engineering-efficacious teachers.

**Keywords:** teacher education; clinical experience; practicums; engineering education; situated learning

## 1. Introduction

It is predicted that many of the jobs of the future will be in the fields of technology and engineering [1] and the USA Bureau of Labor Statistics suggests that modern growth in STEM fields will lead to an excess of 600,000 engineering and engineering technology job openings in the USA alone by 2024 [2]. To address this looming shortage of workers in engineering fields, the workforce of the future will require engineering literate graduates who are engaged in and excited about possible engineering careers. However, many primary and secondary students, who are the future workforce, report they are incapable of becoming engineers because either they do not understand what engineers do or they do not think they have the abilities needed to become an engineer [3]. These findings are especially prevalent for underrepresented groups in engineering. A lack of engineering exposure is one possible cause for these reported deficiencies. Because students' interests in [4] and prior knowledge of a profession [5] are reported to influence career choices, a lack of exposure to engineering in primary and secondary stages could limit the number of students pursuing engineering careers. Further, additional influential factors such as hard-to-fill teacher positions, limited access to professional development, and fewer STEM course offerings could be exacerbating the insufficient exposure to engineering for primary and secondary students [6].

The quality of engineering students in India, China, and the USA has been tied to their educational experiences [7], indicating that strengthening pathways into engineering careers, as well as preparing engineering literate students, requires engineering literate

teachers. Countries such as Australia [8], the United Kingdom [9], and the USA [10] are calling for a national focus on engineering professional development for primary teachers. Research suggests that quality teacher professional development can impact both teachers' perceptions of STEM learning and their confidence to teach it, and also their students' engagement in STEM activities [11]. Conversely, preservice teachers (PSTs) report feeling less prepared to teach STEM due to lower perceived levels of both STEM knowledge and pedagogical content knowledge (PCK) [12]. Hence, the impact that the STEM curriculum can have on student engagement should be recognized, as should the role that teachers' perceptions play in effective implementation. Therefore, re-envisioning teacher training around engineering education is warranted.

To re-envision teacher education, it becomes critical to candidly analyze not just the support structures that result in effective teacher training, but the constraints that limit it too. The realities of teaching during a pandemic placed many preservice teachers in field experience contexts for which they were unprepared, dealing with even the most basic of challenges such as Internet connectivity. Yet according to a 2020 survey, teachers reported issues beyond Internet availability that impacted teaching during COVID-19, including a lack of training in technology tools, increased frustration from parents, an inability to effectively monitor student work, and an associated drop in student engagement [13]. In March to May 2020, US students lost on average 19 days of schooling [14]. New teachers entering their first field experiences in Fall 2021 were asked to accommodate for these learning losses and be prepared to do so both online and in socially distanced learning environments.

## 2. Literature Review

### 2.1. Field Placement

A variety of international policy initiatives have resulted in a global focus on field-based teacher preparation as a primary PST training mechanism [15]. Field placements continue to serve as a crucial component of teacher education and are essential to the construction of PSTs' beliefs about teaching, as well as their capacity for effective teaching [16]. Traditionally, clinical experiences take place at the very end of the teacher preparation sequence, yet programs are now building in frequent and early field opportunities within teacher education programs [17]. Presently, most teacher education programs rely on a series of field experiences that take place over the course of the teacher education program and often rely on a school–university partnership [18]. Although a variety of approaches to preservice teacher education exist, in the traditional model, "college-recommending", PSTs receive coursework on "how" and "what" to teach, and then apply those new understandings in their field experiences [19]. Research on field experiences suggests student teaching provides critical space for PSTs to develop their teaching identities, and science teacher identities in particular, which can result in more personal connections to and confidence in teaching science [20]. Further, field experiences afford a venue to move theory to practice and apply what is learned in university-based coursework [21], as well as opportunities to witness social injustices, practice equitable teaching, and build inclusive learning environments [20].

Coursework and clinical experiences should be complementary, allowing PSTs to connect their coursework learnings and observations during clinical experiences [22]. For example, clinical experiences afford opportunities to develop both content knowledge and pedagogical knowledge [23]. Much of what teachers need to learn about teaching is specific to particular content; in other words, teachers need clinical experiences where they can grapple with the challenges students face when learning particular content [17]. Ultimately, field experiences provide space for PSTs to push their understandings of teaching and learning beyond what they learned from their own experiences as a student, what Lortie [24] calls "apprenticeships of observation" [25].

## 2.2. Engineering Education Professional Development

Primary teachers often report being unfamiliar with engineering content [26,27] and lack experiences teaching it [26]. Teacher preparation programs have not traditionally incorporated instruction on how to teach engineering into their programs. In fact, only 3% of primary teachers in the USA reported having any college coursework in engineering and only 3% reported feeling well prepared to teach engineering to their students [28]. A lack of familiarity with engineering can lead to limited conceptions and misconceptions about the field [29,30] which can influence teachers' approaches to teaching engineering [31]. Further, research has shown that teachers are uncomfortable teaching concepts they are unfamiliar with (Brophy et al., 2008), which may result in them avoiding teaching the subject at all [32].

To address these challenges, teacher educators must provide opportunities for primary teachers to enhance their knowledge and teaching efficacy related to engineering. The Standards for Preparation and Professional Development for Teachers of Engineering [33] stress the importance of providing opportunities for teachers to develop engineering PCK knowledge. PCK refers to a teacher's ability to present content in a way that is easily understood by others [23]. Teachers who possess high levels of PCK understand what makes particular concepts difficult to understand and the preconceptions and misconceptions students have about a subject. Further, they can employ teaching strategies, such as engineering design-based teaching, that make the concepts accessible to the students they are teaching.

Providing opportunities to develop PCK can also provide mastery experiences to enhance engineering teaching efficacy. Teacher efficacy, an extension of Bandura's self-efficacy [34], refers to a teacher's belief in their ability to influence student learning [35] and consists of both general teaching efficacy and personal teaching efficacy [36]. Teacher efficacy varies across contexts such as subject matter and age of the students taught [37] and is a strong indicator of how successful a teacher will be in the classroom [38]. The correlation between teacher efficacy and classroom success has led to researchers using teacher efficacy to explore the impacts of engineering-focused learning opportunities on teachers [39–42]. For example, providing PSTs with opportunities to engage in engineering-focused training and later teach engineering lessons to primary aged students has been shown to enhance engineering teacher efficacy [40,42]. Engineering teacher efficacy has even been shown to increase without direct engineering teaching opportunities when pre-service [39] and in-service primary teachers [41] are provided with authentic opportunities to engage in engineering design challenges as learners. These studies illustrate that primary teachers' efficacy for teaching engineering can be enhanced when they teach engineering activities to their students as well as when they engage in engineering-focused learning opportunities themselves.

## 3. Theoretical Framework

This project integrates Social Cognitive Career Theory (SCCT) and Situated Learning Theory (SLT) to examine preservice elementary teachers' experiences while teaching engineering and the formation of their teacher identities. To prepare effective engineering teachers, teacher educators need to help PSTs enhance their engineering-related teaching efficacy by providing them with mastery experiences occurring within authentic contexts (SLT) that will hopefully bring about career-related affirmation beliefs (SCCT). Consequently, we rooted our study in those frameworks, using SCCT and SLT as a lens to investigate their experiences and teacher identity formation.

### 3.1. Social Cognitive Career Theory (SCCT)

SCCT is a conceptual framework for understanding the aspects involved in career development [43]. SCCT draws from Bandura's social cognitive theory and presents three building blocks of career development: self-efficacy, outcome expectations, and personal goals. Self-efficacy refers to an individual's beliefs about their abilities to succeed. A person's self-efficacy develops in four ways: through personal performance and mastery,

social modeling (vicarious learning from others like you), social support from others, and improvement of psychological and physical well-being [44]. Outcome expectations refer to a person's beliefs about what will result from performing specific behaviors and might include things such as monetary gains, social approval or disapproval, and self-satisfaction [43]. Individuals set personal goals to guide their behavior and increase the likelihood of achieving desired outcomes. According to SCCT, individuals choose careers based in part on their attitudes, values, and interests. Individuals are more likely to have positive attitudes towards and express interests in activities they feel confident in (high self-efficacy) and from which they expect positive outcomes. Additionally, increased interest in a particular activity is likely to result in an individual investing more time participating in that activity. Increased participation or practice in a particular area can also result in improved skills which will lead to higher self-efficacy, thus reinforcing interest in the area [43]. Alternatively, if individuals do not feel confident in certain activities (such as teaching engineering) they will be more likely to avoid participating in those activities.

Self-judged capabilities influence the career options people consider, how much interest they show in a career, and the job paths they ultimately follow [43,45]. According to SCCT, individuals choose career paths based on their interests, attitudes, and values [43]. Because people invest more time participating in activities that they have high self-efficacy in, they are likely to enhance their skills related to those activities, and thus enhance their self-efficacy. Individuals can then choose career paths based upon these developed skills they feel confident in. Conversely, if individuals have low self-efficacy in an area (such as teaching engineering), they may avoid participating in activities that could enhance their skills related to that area.

### 3.2. Situated Learning Theory (SLT)

According to situated learning theory [46], learning is situated, meaning it is embedded within activity, context, and culture. Learning takes place in authentic contexts in which the knowledge would normally be used. For teachers, then, it could be argued that learning how to be a particular type of teacher happens, in part, while the teacher is teaching. Social interaction is an essential component of situated learning as learners become involved in a community of practice that embodies the beliefs that are to be acquired [47]. Teachers are likely to spend less time teaching in a content area in which they have low efficacy [32,48]. Because teachers learn and grow with teaching practice, avoiding teaching experiences due to low efficacy can result in teachers missing valuable learning opportunities that could enhance their PCK [48].

SLT views learning as an identification process in which identities are conceived as "long-term, living relations between persons and their place and participation in communities of practice" [46]. Further, a community of practice is defined as "an activity system about which participants share understandings concerning what they are doing and what that means in their lives and for their communities" [46]. Identity building is a process of negotiating the meanings of one's experiences as a member of a social community [47]. As such, what a teacher knows, does not know, does, or does not do are all negotiated over the course of the job while interacting with others. This creates a unique identity that is shaped by belonging to the community.

Learning is an identification process, and learning (and thus identification) is an "evolving form of membership" [47] within a community of practice. When a teacher encounters unfamiliar content or practices (such as engineering design-based teaching or engineering concepts), the teacher ventures into unfamiliar territory and must engage in legitimate peripheral participation and learn form more experienced peers to move inward from the periphery of the community and assume the role of expert. This apprentice-expert model presented by Wenger [47] assumes that the novice will have access to a more experienced member of the community of practice. Further, the knowledge the novice is learning from the expert must be applied in authentic contexts for the novice to move closer to the center of the community. Having access to an experienced teacher

mentor is an expectation for most field placement students; however, there is a limited number of classrooms in which PSTs can be placed where they will see high quality reform-based teaching practices being implemented [49]. It is not uncommon for primary PSTs to complete their entire teacher preparation program without witnessing high-quality reform-based science teaching or engineering teaching. Studying the ways in which PSTs negotiate their identities as primary engineering teachers within the school and professional communities could have implications for the way scholars view the role of novice/expert in professional identity formation as well as for the ways we approach teacher training, induction, and professional development.

## 4. Purpose

This qualitative case study [50,51] investigates the experiences of two PSTs working in diverse settings—one in a completely virtual classroom and the other in a face-to-face classroom implementing social-distancing protocols. More specifically, this study investigated the experiences of the two primary education preservice educators completing their practicum field experience while confronted with unforeseen circumstances and unfamiliar content. These unforeseen circumstances resulted from an abrupt change in the modality of their teaching experience due to the COVID-19 pandemic, as well as the introduction of unfamiliar STEM content on which a component of their field experience focused.

During the summer prior to their practicum experience, the PSTs participated in a series of professional development opportunities focused on using ethnographic practices and photo journaling in the design and delivery of micro-computer engineering instruction in their practicum primary classrooms. Because both PSTs originally planned to complete their practicum experience within physical classrooms, the training focused on the integration of those strategies and the curriculum in face-to-face contexts specifically. Unfortunately, the resurgence of COVID-19 in late summer 2020 resulted in a shift to hybrid and fully online instruction, and the need to integrate approaches for which the PSTs were not prepared. Consequently, we focused our research efforts on the PSTs' experiences considering that better understanding how the PSTs adapted to the unforeseen circumstances and unfamiliar engineering content could provide teacher educators with insights into how best to prepare new teachers for unexpected challenges when teaching engineering.

However, not enough is known about how PSTs respond to and negotiate unfamiliar content and unexpected circumstances when teaching engineering. Further, little research exists on how PSTs' experiences were altered by the pandemic. Even less research exists on how primary PSTs navigated shifts in teaching modality when teaching engineering in online or hybrid contexts. The purpose of this study was to investigate how PSTs respond to and negotiate unfamiliar content and unexpected circumstances when teaching engineering and how those experiences influenced their identities. The following research questions were developed to guide these efforts: (1) How do preservice teachers navigate unfamiliar contexts when teaching elementary engineering? (2) What do preservice teachers indicate that they learn throughout their field experiences?

## 5. Materials and Methods

### 5.1. Overview

This study focuses on the experiences of two preservice primary education majors completing their practicum field experiences as part of their teacher education program and preparation to become primary teachers. The teacher education program is situated within a large, public land grant university in the Rocky Mountain region and each study participant was assigned a pseudonym for confidentiality. The practicum experience is an 80-h field experience occurring after early field experiences (those associated with general teacher education coursework) and before student teaching, where PSTs gradually take the instructional lead in the classroom. In this teacher education program, PSTs are preparing to teach in primary contexts and must complete two total practicum experiences

before moving on to student teaching. We provided each PST a pre-built engineering curriculum that focused on the integration of computer science, electrical engineering, and agriculture science to build soil moisture sensors and automatic watering systems. Given the role that ranching and farming plays in the area, this particular engineering curriculum was selected because the agriculture connection provided the most relevance to the greatest number of participating students. It should be noted that the treatment of engineering disciplines within the pre-built engineering curriculum was performed in adherence to the current literature on K-5 engineering. Consequently, the engineering curriculum did not distinguish among the different engineering disciplines. Instead, the curriculum addressed engineering more broadly, focusing on concepts such as awareness and appreciation of modern infrastructure, and associating engineering with pro-social, community, and sustainability opportunities.

### 5.2. Participants

The first participant, Kristina, is a 4th year student at the university working toward her primary education degree with a science education option. Upon graduation, Kristina will be recommended for licensure to teach US grades K-8 (5- to 14-year-olds). Kristina was placed for her second practicum experience in a local 4th grade classroom (9- to 10-year-olds) where she was eager to apply her training in primary and science education. Her cooperating teacher, or the supervising teacher in whose classroom Kristina was completing her second practicum experience, was Holly. Holly is a 14-year teaching veteran with National Board Teacher Certification, a national level advanced teaching credential. Kristina hopes to teach upper primary or middle/junior high school science when she graduates.

The second participant in this study, Jennifer, is a 3rd year student in the teacher education program who switched majors from engineering to primary education her first year at the university. Although Jennifer was originally an engineering major, she only completed prerequisite coursework for the major and did not complete any engineering-specific coursework. Jennifer is working toward both the science and mathematics education options within the teacher education program. Jennifer was placed in a small rural 3rd grade classroom (8- to 9-year-olds) to complete her first of two total practicum experiences and is hoping to teach middle/junior high school science or math when she graduates. Jennifer's cooperating teacher is Kerri, a 3rd grade teacher with almost 20 years of teaching experience.

### 5.3. Study Design

We employed a nested qualitative case study approach [51,52] to explore PSTs' experiences during a field placement where they were confronted with both unforeseen circumstances and unfamiliar content. The case or "bounded system" [51] of interest in the current study is a university practicum field experience occurring during the COVID-19 pandemic. Nested within this single case are two individual participants, each constituting their own case. We employed purposeful intensity sampling [50] to identify two participants who met the following criteria: (1) their field placement experience consisted of unexpected circumstances beyond what might typically be expected, and (2) they were in charge of delivering unfamiliar content to elementary students in the form of technology-enhanced engineering curriculum. Intensity sampling allowed us to identify participants whose experiences were unusual enough to provide information-rich examples that could illuminate both the typical and atypical.

### 5.4. Data Collection

We collected multiple forms of data, focusing on the three data collection techniques Merriam describes as the "most appropriate for qualitative case study research" [51]. These included participant interviews, video observations, and documents—reflective teaching

journals in the present study. The use of multiple data sources provided diverse material for analysis and allowed for data triangulation [50].

### 5.4.1. Interviews

We conducted one in-depth interview with each preservice teacher within two weeks after they finished their practicum experience. Each interview spanned approximately 1.5 h and was video recorded and transcribed. We utilized a combination interview approach making use of both a standardized open-ended interview protocol and an informal conversation [52] to "capture the perspectives of program participants" [52]. This combined approach ensured we were collecting the same data from both PSTs, while simultaneously providing flexibility to explore additional topics that an individual interviewee deemed important. The interview questions were developed to elicit information around (a) participants' prior training to prepare them for the experience (e.g., "What facilitation/guidance did you receive for teaching in the modality that you did?"), (b) the field placement experience itself (e.g., "What was your role in the classroom?", "What sorts of conversations transpired between you and your cooperating teacher? Talk about the topics and highlight some of the most interesting conversations you had."), (c) experiences teaching engineering (e.g., "How did you integrate lessons learned from the summer professional development into your teaching of engineering?", "Can you share more about your experiences with teaching engineering?"), and (d) thoughts on their own learning (e.g., "What did you learn about yourself from this experience? As a teacher? As a learner? As an engineering literate person?"). The interview protocol was shared with the PSTs two days before the interview to allow them time to organize their thoughts around the questions.

### 5.4.2. Reflective Journals

Throughout the field placement experience, PSTs were asked to take ethnographic field notes using the notetaking and notemaking approach described by [53]. The participants composed the field notes following each individual classroom experience in which they led the engineering activities. Although the PSTs were not required to use a template to construct their field notes, each of their notes consisted of detailed and substantive observations as well as their personal reflections on what they observed and experienced while engaging in field placement activities (e.g., observing mentor teacher, lesson planning, lesson delivery). Each PST also included students' reactions to the lessons with exemplar quotes, as well as steps they would address if they were to re-teach the lesson. Further, each of the participants' field notes were equally as substantive and detailed in nature.

### *5.5. Data Analysis*

Data collection and analysis occurred simultaneously, with each author creating analytic memos throughout the process [54]. After all transcripts and documents were collected, formal data analysis began and consisted of multiple coding phases and collaborative sense-making sessions. The process began with the three of us independently open coding [54] the data, first by lumping the data into chunks and assigning initial codes, and then going back through each chunk to complete line by line coding. After completing this initial round of open coding, we came together for a collaborative sense-making session. During this session, we discussed the results of our individual coding and developed a shared codebook to use during a second round of coding. In addition to codes that emerged from the open coding process, the codebook contains a set of codes connected to the theoretical frameworks that inform this study. Table 1 provides an excerpt from the resulting codebook.

**Table 1.** Excerpt from codebook.

| Code | Description | Examples |
|---|---|---|
| Challenges | Challenges refer to those challenges that hindered the design and delivery of the curriculum or to challenges with the curricular materials or school infrastructure. | "They have to wear masks and like wash their hands and hand sanitizer all the time. Um, and I definitely struggled with it and so this year was super different just in having kids, [physically distanced] as much as possible" (Jennifer interview) |
| Confidence and Self-Efficacy | Confidence and self-efficacy refers to preservice teacher's belief in their ability to influence student learning [34,35]. | "You really don't need to know everything about engineering to teach engineering. My confidence with teaching engineering I think was the biggest thing that had changed because as a teacher you just it's impossible to know all you need to know." (Kristina interview) |
| Pedagogical Growth | Pedagogical growth refers to the preservice teachers' development of their knowledge of teaching and learning. | "Some kids maybe we're coding or they're doing extension and coding even more and some kids are focusing on identifying the problem and some students are doing the building part and, again, still student centered and I'm going around asking questions. But I think the difference here is that it's. There's a lot going on. Moving to the same goal but the students are at different parts of the same goal" (Kristina interview) |
| Professional Growth | Professional growth refers to the preservice teachers' development of their understandings and perceptions of the professional dimensions and responsibilities about teaching. | "This portion of the lesson may have been rushed because I didn't explicitly think about how I was going to connect it (what questions I would ask students, how I would have the students discuss this, etc.). A consequence of this is that students weren't able to fully connect their own personal experiences to the lesson. Had they made this connection, they would have been more intrigued with the challenge." (Jennifer reflective journal 4) |

After the second round of coding, we met for another collaborative sense-making session to discuss the codes and critical events we had independently identified using the shared codebook. At this point in time, we began a series of three additional collaborative sense-making sessions to jointly engage in axial coding [54]. During the axial coding sessions, we strategically reassembled the data and identified the most important categories and associated categories present in the codes. Through the axial coding process, the interconnectedness of the resulting themes was identified.

*5.6. Trustworthiness*

Throughout data collection and analysis, a number of Lincoln and Guba's [55] techniques for establishing trustworthiness were employed by the research team. First, data collection occurred over an entire academic year, and we engaged in a prolonged period of establishing rapport with participants [56]. Secondly, an audit trail has been carefully maintained to ensure full transparency in the decisions we made about our interpretations and analyses. This included maintaining the integrity of all raw data as well as notes documenting our data reduction process and decisions. We engaged in multiple methods of triangulation [51,52] including the use of multiple investigators for collecting and analyzing the data; the use of multiple theories to confirm emerging findings; and the use of multiple data collection methods to establish confirmability, including cross-walking between the participants' field notes and interview data. Lastly, we built rich, thick descriptions of the participants' experiences, and used those descriptions to member check with the PSTs.

## 6. Results

Following data analysis, we composed case summaries and then framed responses to each of our research questions. Case summaries, accompanied by a case summary table, are included along with responses to the individual research questions.

### 6.1. Kristina: "I Found My Passion"

Although the school at which Kristina was placed opted to return to in-person learning in the fall, Kristina's cooperating teacher was selected to teach online for those 4th graders who were not able nor comfortable returning to face-to-face instruction. Kristina had not planned on completing her second practicum in an online and virtual context, but as expected, tackled the challenge with enthusiasm and excitement. Her cooperating teacher, Holly, is considered by many to be a technology leader in the school and across the district, and Kristina was eager to experience online teaching and learning under her guidance.

Kristina shared on several different occasions how much this experience impacted her thinking about science instruction and shared some notable impacts on her perceptions of engineering and how to best teach STEM content. Most importantly, Kristina's confidence in teaching engineering increased considerably from the start of the project to the end. She shared that prior to this experience, "engineering before was so distant to me and I was detached from it; it was a different world", but after the experience, she "become more confident in my understanding of engineering" through the opportunity to teach it. Although she was at first concerned about teaching STEM, especially under COVID-19 constraints, she soon found passion for both teaching and learning STEM material. She said, "I was so excited and had so much motivation throughout the whole thing like once I got into it, I was like, Yes, this is what I want to do." Because the engineering curriculum she delivered included considerable focus on computer science, and more specifically on coding, a content area that she found daunting yet intriguing, she was exposed to a STEM discipline that ignited her passion and validation for her chosen career. This experience afforded a series of learning opportunities that ultimately resulted in the type of career affirmation event we want all PSTs to experience during their teacher education trajectory. She shared that after the experience, "I hope to be a STEM teacher, you know in middle school or elementary school, and help implement this sort of stuff and moving forward in whatever district I'm in". The experience also afforded Kristina the opportunity to explore the true power of collegial and collaborative relationships during her time in practicum. Holly routinely revisited the best-practice research in teaching with Kristina, reminding her that it is unrealistic to know everything as an educator. Kristina shared, "Going into this experience, I thought I needed to know everything about STEM in order to teach it, but I actually learned a lot along with the kids." The case summary for Kristina is provided in Table 2.

**Table 2.** Summary table for Kristina.

| Affordances | Constraints | Outcomes |
|---|---|---|
| Already had one teaching field experience before joining the project<br>Prior experience as a student in classes taught by the project leads<br>Cooperating teacher was considered a technological leader in the district<br>Taught remote instruction alongside her cooperating teacher in an empty classroom<br>Was given freedom to design lesson plans and incorporate differentiation and scaffolding techniques | Limited prior experience with engineering concepts and skills<br>Entered the year with little belief that she would teach for a career<br>School opted for online classes for the duration of the school year<br>Asynchronous class format led to limited opportunities for formative assessment and student noticing | Found critical support and guidance from her mentor teacher during one-on-one time<br>Confidence in STEM content knowledge increased throughout the project<br>Discovered an interest in Computer Science<br>Grew a deeper appreciation for best pedagogical practices based in research<br>Appreciated the need to always be learning and growing as a teacher |
| | | **Key Outcome** |
| | | The experience reignited Kristina's passion to teach elementary or middle school STEM |

### 6.2. Jennifer: The Power of Consistency

Jennifer was paired up with Kerri, a veteran 3rd grade teacher with almost 20 years of teaching experience. When we spoke to Kerri's administrator about possible participants in our project the summer before, Kerri was immediately suggested. Kerri has extensive experience working with PSTs, as well as a keen interest in STEM learning, making her a perfect fit for a cooperating teacher under which Jennifer could train. Unlike Kristina, the school at which Jennifer was placed decided to return to full in-person learning in the fall at the start of her practicum experience. It should be noted that this was the first of two practicum experiences for Jennifer, and the first practicum experience is often a teacher education student's first substantive dive into the work of K-12 education. Jennifer shared that the 17 students in her class were very successful in negotiating COVID-19 constraints throughout her experience, including masking, washing hands, and social-distancing. Under normal circumstances, Jennifer's cooperating teacher encouraged flexible seating. Jennifer, however, shared that it took some getting used to for everyone, herself included, to manage the impact of constraints such as social distancing on elements of teaching such as group work. Further, Jennifer was faced with a series of extenuating circumstances that constrained her experience and resulted in her taking the primary lead on science instruction. Most notably, Kerri, the cooperating teacher, encountered a health issue midway through Jennifer's placement, resulting in the need for her step away from the classroom. This left Jennifer largely in charge of science instruction.

During her time delivering the engineering curriculum while Kerri was still in the classroom, Jennifer was able to lean into the strong relationship she developed with Kerri. Like Kristina's experience with Holly, Jennifer was afforded Kerri's expert guidance and mentorship which she reported was critical to her successes during the practicum experience, most notably her capacity for recognizing the needs of her students. Jennifer said that Kerri "understood the needs of our students, and it was really great just talking with her like one on one, about what those needs are". Further, this resulted in learning opportunities for both her and her learners. She saw both her own perceptions of engineering, as well as her students' perceptions of engineering, make a considerable shift. Jennifer shared with us that she witnessed her students' ideas move from considering engineering as a physical manifestation of engineering to engineering being about problem solving. She said she observed "their view of engineering expand to problem solvers and not just you know like cars and bridges and cell phones". She was also delighted that her students were able to see beyond the agriculture-focus of our engineering curriculum and apply the concepts addressed into other STEM realms.

From a teacher and learning perspective, Jennifer quickly learned the importance of having a back-up plan as a teacher in case you suddenly "don't have access to internet or computer", and the need for flexibility and the ability to pivot when things do not go as expected. The experience in Kerri's class also provided Jennifer with a learning opportunity to understand the critical roles of scaffolding and differentiation in lesson design, two teaching practices we know often challenge our PSTs. She said she was grateful for the experience to explore "how can we support the struggling students and how can we support the advanced students." Ultimately, like Kristina, Jennifer left the experience with a deep sense of career affirmation. More specifically, this primary school-focused field experience afforded her the chance to confirm that she prefers to work with older students and is excited about her future as a middle school educator. She said, "Always thought I'd teach middle school, and I still think that that's probably right for me." Most importantly, she was certain that her love of science and engineering would continue to evolve into a cornerstone of her identity as an educator. In summing up her experience, she shared, "What I learned about myself, is one I really, I love science and engineering." Table 3 provides a summary table for Jennifer.

**Table 3.** Summary table for Jennifer.

| Affordances | Constraints | Outcomes |
|---|---|---|
| Pursuing math/science teaching option with an interest in STEM<br>Prior experience as a student in courses taught by the project leads<br>Paired with a veteran 3rd grade teacher with STEM experience<br>School returned to full in-person learning<br>Rural district presented opportunities to connect agriculture sciences with the STEM curriculum<br>Was given freedom to design lesson plans and incorporate differentiation and scaffolding techniques | First substantive experience within K-12 Education<br>COVID-19 protocols forced masking and social distancing within the classroom<br>Stepped in as the primary lead on science education in response to a medical emergency<br>Limited opportunities to seek support from her cooperating teacher following the medical diagnosis | Found critical support and guidance from her mentor teacher early on<br>Noticed her and her students' perceptions of engineering expand to more authentic contexts<br>Learned to have a back-up plan for when technology didn't function as expected<br>Recognized the power of consistency to positively impact teaching and learning<br>Recognized the importance of growing from mistakes in pursuit of learning<br>Affirmed her belief that she wants to teach middle school science into the future |
| | | **Key Outcome** |
| | | The experience cemented Jennifer's love for both learning and teaching STEM concepts |

### 6.3. Research Question #1: How Do Preservice Teachers Navigate Unfamiliar Contexts When Teaching Primary Engineering?

Our first research question focused on examining how the PSTs navigated the unfamiliar contexts with which they were presented. In examining this question, findings suggest the contextual factors that shaped each participants' field experience resulted in affordances and constrains with which the PSTs had to negotiate. It was contending with these affordances and constraints that ultimately provided learning opportunities through which each of the PSTs grew pedagogically and professionally. More specifically, the contextual factors that framed Jennifer and Kristina's practicum teaching experience required them to negotiate the affordances and constraints, and this act of negotiating the affordances and constraints eventually led to learning opportunities for each PST and their growth as educators. We have purposefully chosen the term "negotiate" to represent the process by which the PSTs mitigated the constraints as best as they could, while maximizing their return from the affordances the unfamiliar contexts brought, and we have operationalized the terms "affordances" and "constraints" from the literature based on usability and interface design. Affordances are the attributes of the context and the relationships within it [57] that make things possible, and constraints are those properties that restrict those possibilities [58,59].

#### 6.3.1. Affordances

Although Jennifer and Kristina were each completing their practicum experience and were participants in this study delivering the engineering curriculum, each was presented with their own set of unique contextual factors that required them to negotiate the affordances and constraints they encountered. First, Jennifer experienced several affordances that allowed her to maximize her experience in the classroom. Jennifer was pursuing both the math and science options to accompany her elementary education degree and license, meaning that math and science were not only of pedagogical interest to Jennifer, but that she had completed additional courses equipping her with content knowledge many of her peers did not have. Additionally, her cooperating teacher Kerri was a veteran teacher with considerable STEM experience and known by her colleagues as a STEM and technology education expert. Unlike Kristina, Jennifer's school returned to full in-person learning at the start of her practicum experience, giving her a practicum experience much more aligned with the training she had received in the teacher education program which, prior to the pandemic, focused on preparing teachers to teach in person.

Jennifer also was given considerable latitude to fully design the engineering lesson plans and integrate pedagogical skills she had been learning about in her coursework, such as the role of differentiation and scaffolding. Lastly, the rural context in which Jennifer was placed presented her with multiple opportunities to connect the engineering curriculum to the agricultural nature of the local community's economy and identity.

Like Jennifer, Kristina encountered a series of affordances within the practicum context that supported her growth. First, she had worked with us during previous coursework, resulting in a familiarity and comfort with the project leads. Additionally, this was her second practicum experience, and she was able to leverage the lessons learned during her first substantive experience in the classroom during her first practicum. Kristina's cooperating teacher, Holly, was a National Board Certified Teacher, and held a master's degree in educational technology. Consequently, Holly was considered by her colleagues within the district to be an expert teacher and mentor. Although the online modality in which Kristina was teaching constrained her experience, it also meant that she had unlimited access to Holly as a mentor. Kristina took full advantage of this affordance and used every opportunity to work closely with Holly to build her professional practice. Lastly, Kristina was given full latitude to adapt the engineering curriculum, and integrate pedagogical practices explored in her course work, such as scaffolding and differentiation.

### 6.3.2. Constraints

Despite these affordances each PST experienced, they were presented with a number of constraints that resulted in unforeseen challenges and hurdles with which they had to contend. Jennifer was completing her first of two practicum experiences, which meant that this was her first in-depth experience in a classroom. Although she had completed other early field experiences, this was the first where she was fully engaged in legitimate peripheral participation and ultimately shifting inward from the periphery toward the role of expert in the classroom [47]. Like Kristina, the pandemic presented further constraints that Jennifer reported as having influenced her experiences. She shared that COVID-19 protocols such as social-distancing prevented her from experiencing the very successful use of flexible seating that had become a hallmark of Kerri's teaching and management of the classroom learning environment (see quote in Table 1 challenges column). Jennifer also reported that masking made it difficult for her to hear her students, and for her students to hear her at times. Further, Jennifer described being constrained by technical challenges. This included complex technical challenges around things such as Internet connectivity and coding of the microcontrollers in the engineering curriculum, to more mundane but equally burdensome constraints such as storage of classroom materials. Jennifer also reported pedagogical constraints. She felt unprepared for the need to scaffold the lessons due to their complexity at times, and that managing the classroom learning environment took considerably more time than anticipated. The most considerable constraint Jennifer faced was when Kerri developed a long-term medical issue that required her to miss consecutive days each week for treatment. This meant that Jennifer ended up with limited access to Kerri's mentorship over the course of the semester and practicum experience. Although Jennifer reported having a very strong university field supervisor, she still craved the support one could get from an expert mentor such as Kerri.

Kristina also experienced a series of constraints that could have limited her experience. Unlike Jennifer who was previously an engineering major and was pursuing a science and math teaching option, Kristina had little experience with engineering education. More specifically, she shared that her limited conceptions of engineering restrained her initial capacity to introduce the curriculum to the learners. She was able to quickly overcome these deficiencies, but she was quick to remark on the initial impact it had on her planning and instruction. Further constraining her experience was her initial mindset about her career choices at the start of her second practicum experience. At the time, she foresaw little interest in entering the teaching profession upon graduation. Jennifer also experienced technical challenges throughout the experience, mostly with the hardware and coding

platform that was central to the engineering curriculum. She also experienced pedagogical challenges such as pacing and a lack of PCK in engineering. This resulted in a sense of initial low self-efficacy in both engineering and coding, and the desire for stronger differentiation skills. The teaching modality also presented possible constraints to Jennifer. As shared, she was suddenly shifted to having an all-online practicum experience just as the school year started, and she felt her teacher preparation coursework had left her woefully underprepared to be teaching in a fully remote context. Eager to apply those skills and concepts she had learned in her coursework, she immediately became aware that her ability to formatively assess student learning was considerably hindered. For example, the asynchronous nature of her instruction led to limited opportunities to witness student thinking, and gauge who had completed work and who had not. More importantly, she said, "It was hard online because we couldn't be there and see exactly how they were responding". This resonates with previous research about the impact of the COVID-19 pandemic on teachers' capacity to monitor student progress [13]. This separation of time and space was particularly challenging when it came to her use of formative assessment to make the learning more meaningful and relevant to her students. She said if she were to do it all over again, she "would try to be more in tune with their personal experience and, you know, asking more questions about how could this be, you know, how do you see this in your individual life and connecting it to their personal experience."

*6.4. Research Question #2: What Do Preservice Teachers Indicate They Learn throughout Their Field Experiences?*

Our second question focused more precisely on what the PSTs learned during their practicum experience teaching engineering. Our findings indicate that it was the negotiation of the affordances and the constraints that shaped each participants' field experience. This negotiation is what led to the learning opportunities each preservice teacher experienced and ultimately their pedagogical and professional growth.

6.4.1. Jennifer

For Jennifer, it was the negotiation of these affordances and constraints that resulted in a series of outcomes that have ultimately shaped her identity as an educator. Prior to Kerri's medical leave, Jennifer was able to forge a critical mentor/mentee relationship with her early in the experience. Despite facing her own considerable challenges, Kerri continued to provide critical classroom support as well as possible during her leave. Jennifer noted the importance of this support, and now recognizes the role that being part of a community of practice with access to mentors can play in her professional growth. Negotiating the unfamiliar engineering content resulted in content knowledge growth for both Jennifer and her students. She reported that their perceptions of engineering had clearly expanded to include more authentic contexts. It also meant that she had to build her PCK in teaching engineering. The negotiation between the affordances and constraints resulted in pedagogical knowledge growth, too. She shared that she learned the importance of having a back-up plan when designing instruction to deal with the unexpected challenges that routinely arise when teaching. In the end, she was able to recognize the critical importance of differentiation, scaffolding, and student-centered learning. She also recognized how important consistency was to her training and her students' growth. She shared that, for both her and her students, the engineering curriculum the students were experiencing was the most consistent component of their school year that was upended by COVID-19 and Kerri's illness. Finally, the negotiation within the unfamiliar context results in professional growth too. She demonstrated a newly discovered sense of growing from one's mistakes in the classroom. When discussing a challenging lesson Jennifer felt did not go as planned, she shared, "This portion of the lesson may have been rushed because I didn't explicitly think about how I was going to connect it (what questions I would ask students, how I would have the students discuss this, etc.)." Not only was she able to critically reflect on what she could have done better, but also recognize adverse impacts of the rushed

lesson on students' learning. She said, "A consequence of this is that students weren't able to fully connect their own personal experiences to the lesson." Given a focus of the engineering curriculum was on building personal connections between engineering and students' identity, this was a particularly insightful remark. Jennifer also reported that the experience teaching 3rd grade affirmed her belief that she would prefer to teach middle school, and not 3rd grade, and that her choice to focus on STEM instruction with the science and math options was the correct one. Finally, the experience negotiating the affordances and constraints in the unfamiliar context seemed to cement her love for teaching and learning, and more importantly, teaching and learning STEM. She was so moved by the experience, she continued to remain involved with her practicum classroom, delivering the engineering curriculum well after her teacher preparation coursework requirements had ended, suggesting the meaningful impact of the relationships she had built with the learners and the deep connection with the content that had developed. At the same time, the negotiation also led her to be very transparent about her recognition of the role that lifelong learning will play in her professional development, suggesting, "I have a lot more to learn".

### 6.4.2. Kristina

Negotiating with these affordances and constraints resulted in a series of outcomes that presented learning opportunities for Kristina. First, because of the pandemic constraints and shift to online learning, Kristina was given the unique opportunity to spend unrestrained one-on-one time with Holly in a mentor/mentee arrangement that few PSTs ever get to experience. This also meant she was able to routinely discuss research-based best practice pedagogy with Holly, and a greater appreciation for rooting her practice in research emerged. Because of this close mentoring relationships, Kristina was able to experience components of good teaching, such as differentiation, scaffolding, and student-centered learning. For example, one of her students was a member of Jehovah's Witnesses. Because of this, Jennifer and Holly worked to differentiate to accommodate that student's political views and understandings of science. She said, "I'm going to see stuff like that as a teacher based on religion or, you know, beliefs of the family. Political views, I mean science really gets into a lot of that" and recognition that in her future practice, she will have to "adapt a lot of different things based off of that." When asked specifically about differentiation, she shared that she came to the realization that differentiation is critical because it means that everyone is, " . . . Moving to the same goal but the students are at different parts of the same goal." The negotiation of the context also provided the unique opportunity to co-learn with her students. This gave her a chance to model lifelong learning and critical dispositions, such as being comfortable with safe-to-fail learning contexts. Further, although initially constrained by her lack of experience with engineering, Kristina's passion and confidence in STEM, and engineering in particular, gradually increased over the course of the practicum experience. In terms of her growth in confidence, Kristina shared, "You really don't need to know everything about engineering to teach engineering. My confidence with teaching engineering I think was the biggest thing that had changed because as a teacher you just it's impossible to know all you need to know." While negotiating the affordances presented by the autonomy she was given to adapt the engineering curriculum and the constraints presented by her inexperience with engineering, she was able to kindle a new interest in both computer science and coding, as well as teaching about energy. Combined, these negotiations resulted in a career-affirming event and re-ignition of a seemingly lost interest in teaching either elementary or middle school STEM education. Because of the experience, Kristina was again thrilled about her future career as a STEM educator, and like Jennifer, she was left with a promising sense of the importance of being a lifelong learner as an educator, saying, "I have a lot to learn".

### 7. Discussion

The purpose of this study was to investigate how PSTs respond to and negotiate unfamiliar content and unexpected circumstances when teaching engineering. Data analysis indicates the PSTs seemingly adopted unique tactics to delivering the engineering intervention, resulting in diverse approaches to implementing professional development activities and engineering curriculum into their classroom practice.

Two primary themes emerged from this study. First, field experience contextual factors both afford and constrain learning opportunities for PSTs. More specifically, the contextual factors each PST faced shaped their field experience and resulted in affordances and constrains with which the PSTs had to contend. It is important to note that both affordances and constraints were found in the influencing contextual factors. In other words, there were contextual attributes that constrained, or limited, the PSTs' experience and growth such as the social distancing requirements due to the pandemic, the PSTs' lack of engineering PCK, or the PSTs' experience with differentiation. At the same time, there were contextual affordances that made learning possible such as the mentorship the PSTs received, prior experience with STEM teaching and learning, and autonomy when implementing the engineering education curriculum. Secondly, it is the negotiation of these affordances and the constraints that result in PSTs' pedagogical and professional growth. Contending with the affordances and constraints each preservice teacher faced is what ultimately resulted in the learning opportunities that led to their growth. The PSTs needed to mitigate the limiting factors of the constraints and take full advantage of the affordances with which they were presented. For example, Kristina had limited experience teaching engineering, but was given considerable autonomy to adapt the engineering curriculum to emphasize energy instruction, something with which she was much more comfortable. This negotiation gave her the space to ultimately build her engineering pedagogical knowledge and engineering teaching self-efficacy.

This study is important because it provides critical insight into the experiences of two PSTs who had to negotiate unfamiliar content and unexpected circumstances while teaching engineering. From this, we have taken two primary implications from our findings. First, findings can be used to help teacher educators better structure how to more effectively prepare PSTs to encounter unforeseen circumstances during their clinical experiences. For example, teacher educators could help PSTs categorize contextual factors into affordances and constraints and facilitate the process of negotiating those factors. Further, teacher educators need to help PSTs realize that dealing with affordances and constraints, and the negotiation process, is active and ongoing. That process is what leads to professional and pedagogical growth. Teacher educators must keep in mind that PSTs are novices, and as a result, they are not going to know that they must tackle those affordances and constraints and learn how to negotiate them. We consider the PSTs in this study as exceptional teacher candidates who were largely able to tackle this process on their own. We also realize, however, that not all PSTs will have the capacity for this negotiation process. Consequently, teacher educators could better frame analysis of and reflection on those affordances and constraints and how best to negotiate them to arrive at the most effective clinical experience. To do this, teacher educators could help PSTs identify affordances and constraints, and then strategize with them to support negotiation, resulting in pedagogical and professional growth.

Secondly, findings from this study can be used to better understand how best to prepare engineering-efficacious preservice teachers. With previous research suggesting over 600,000 engineering job openings by the 2024 in just the United States [2], it could be argued that primary and secondary education needs to revisit how best to expose students to engineering. Thus, more exposure to engineering for primary and secondary students could lead to more effectively building their interest in engineering [4] and preparing them to become engaged with and seek engineering careers. However, developing primary and secondary students' engineering literacy and interest in engineering careers will require engineering literate teachers. Yet primary PSTs often lack both engineering content knowl-

edge and experience teaching it [26,27]. Providing opportunities to experience engineering education in the authentic space of a field experience might be one way to build PSTs' engineering PCK [33] and build their engineering teaching self-efficacy [40,42]. Clinical experiences that have an emphasis on engineering education could provide PSTs the critical opportunity needed to better teach engineering and build engineering PCK [17]. Further, clinical experiences where PSTs can engage in engineering education might serve as a space for them to develop their teacher and science teacher identities, resulting in a stronger personal connection to engineering and engineering self-efficacy [20]. In turn, this increased engineering self-efficacy might mean the PST develops into an more successful educator in the classroom [38], ready to appropriately teach engineering and build engineering interest and capacity, especially in primary learning contexts. This emphasis on primary contexts is in direct response to international calls to action for focus on engineering professional development for primary teachers [8–10].

## 8. Conclusions

Findings from this study indicate that PSTs face contextual factors that both afford and constrain learning opportunities. This theme is important because it recognizes that even the constraints faced during a field experience can be navigated by the PST to arrive at a learning opportunity. In many ways, it would be hard to challenge the idea that those contextual affordances, or the factors in a field experience that make things possible, would do anything but result in growth for the preservice teacher, but what about the constraints, or those factors that seemingly limit the experience for the preservice teacher? Our study shows that the challenges presented in field experiences when teaching engineering can also give rise to professional and pedagogical growth. The PSTs found ways to mitigate those constraints and leverage them toward their own growth by working within and around the limitations and affordances. This study has provided insight into this process of negotiation. Better understanding what contextual factors might be considered affordances and which ones might be considered constraints can illuminate how to streamline clinical experiences and facilitate the construction of PSTs' beliefs about teaching and their capacity for effective teaching [16] Ultimately, findings from this study could be used by teacher educators to construct examples on how best to prepare teachers to respond to and negotiate unfamiliar content and unexpected circumstances when teaching engineering, especially in primary contexts.

**Author Contributions:** Conceptualization, R.H. and N.L.; Investigation: R.H. and N.L.; Formal Analysis: R.H., N.L. and B.W.; Writing—Original Draft: R.H., N.L. and B.W.; Writing—Review & Editing: R.H., N.L., B.W. and P.G. All authors have read and agreed to the published version of the manuscript.

**Funding:** This material is based upon work supported by the National Science Foundation under Grant No. 1916673. Any opinions, findings, and conclusions or recommendations expressed in this material are those of the author(s) and do not necessarily reflect the views of the National Science Foundation.

**Institutional Review Board Statement:** The study was conducted in accordance with the Declaration of Helsinki and approved by the Institutional Review Board of Montana State University (protocol code RH080119) for studies involving humans.

**Informed Consent Statement:** Informed consent was obtained from all subjects involved in the study.

**Data Availability Statement:** Consistent with the approved ethics, data is not available for general access.

**Conflicts of Interest:** The authors declare no conflict of interest. The funders had no role in the design of the study; in the collection, analyses, or interpretation of data; in the writing of the manuscript; or in the decision to publish the results.

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
