# Peer review of "Building Primary Preservice Teachers’ Identity as Engineering Educators"

_education, doi:10.3390/educsci12100637_

Round 1
Reviewer 1 Report
This manuscript is well written and the detail of the study well presented. I was left with only a few questions including understanding a little more about Jennifer's (the former engineering major) college experience in her engineering field - was this field pertinent to the project students performed and/or did Jennifer take any "real" engineering course work her first year (rather than support courses / prerequisites for her engineering course work)? As always with STEM education studies, I wish the authors differentiated the types of engineering fields that exist rather than lumping things together. That said, the current treatment of STEM in this manuscript is fairly standard.
There was an abrupt change in reference style from APA to numbers about 1/3 of the way through the manuscript and this needs to be addressed.
Reviewer 2 Report
This qualitative study provides insight into the affordances and constraints provided to preservice teachers who not only had to teach engineering content but also had to teach in difficult circumstances during the pandemic. The paper provides a solid background of literature and theory to situate the study. Overall I think the paper is well written and researched. I just had a few comments.
As I was reading, I wondered how the video observations were used in the results. The results seemed focused on the perspective of the preservice teacher (that appeared to use quotes from either interview or reflective journals). Were the video observations used, and if so, how were they used in the results? Were they coded the same as the interviews and journals?
I think it would be helpful in terms of interpreting results to know more about how often the reflective journaling took place for each participant; or whether one participant journaled more often than another.
There were a few minor grammatical errors: lines 377, 476, 534, 543, 673, 724.
